# Omics and CRISPR-Cas9 Approaches for Molecular Insight, Functional Gene Analysis, and Stress Tolerance Development in Crops

**DOI:** 10.3390/ijms22031292

**Published:** 2021-01-28

**Authors:** Muhammad Khuram Razzaq, Muqadas Aleem, Shahid Mansoor, Mueen Alam Khan, Saeed Rauf, Shahid Iqbal, Kadambot H. M. Siddique

**Affiliations:** 1Soybean Research Institute, National Center for Soybean Improvement, Nanjing Agricultural University, Nanjing 210095, China; khuram.uos@gmail.com (M.K.R.); muqadasaleem@gmail.com (M.A.); 2National Institute for Biotechnology and Genetic Engineering, Faisalabad 38000, Pakistan; shahidmansoor7@gmail.com; 3Department of Plant Breeding and Genetics, Faculty of Agriculture and Environment, The Islamia University of Bahawalpur, Punjab 63100, Pakistan; mueen.alam@iub.edu.pk; 4Department of Plant Breeding and Genetics, College of Agriculture, University of Sargodha, Sargodha 40100, Pakistan; saeedbreeder@hotmail.com; 5Laboratory of Fruit Tree Biotechnology, College of Horticulture, Nanjing Agricultural University, Nanjing 210095, China; 2017204045@njau.edu.cn; 6The UWA Institute of Agriculture, The University of Western Australia, Perth, WA 6001, Australia

**Keywords:** plant stress, abiotic stress, biotic stress, omics, CRISPR-Cas9, crop stress tolerance

## Abstract

Plants are regularly exposed to biotic and abiotic stresses that adversely affect agricultural production. Omics has gained momentum in the last two decades, fueled by statistical methodologies, computational capabilities, mass spectrometry, nucleic-acid sequencing, and peptide-sequencing platforms. Functional genomics—especially metabolomics, transcriptomics, and proteomics—have contributed substantially to plant molecular responses to stress. Recent progress in reverse and forward genetics approaches have mediated high-throughput techniques for identifying stress-related genes. Furthermore, web-based genetic databases have mediated bioinformatics techniques for detecting families of stress-tolerant genes. Gene ontology (GO) databases provide information on the gene product’s functional features and help with the computational estimation of gene function. Functional omics data from multiple platforms are useful for positional cloning. Stress-tolerant plants have been engineered using stress response genes, regulatory networks, and pathways. The genome-editing tool, CRISPR-Cas9, reveals the functional features of several parts of the plant genome. Current developments in CRISPR, such as *de novo* meristem induction genome-engineering in dicots and temperature-tolerant LbCas12a/CRISPR, enable greater DNA insertion precision. This review discusses functional omics for molecular insight and CRISPR-Cas9-based validation of gene function in crop plants. Omics and CRISPR-Cas9 are expected to garner knowledge on molecular systems and gene function and stress-tolerant crop production.

## 1. Introduction

Abiotic stresses, such as drought, salinity, temperature extremes, and climate change, are major considerations for scientists. The development of high-yielding varieties exposed to stress depends on direct selection for yield stability in multiple locations. Germplasm development with tolerance to biotic and abiotic factors is important for sustainable crop production [1,2]. The molecular term “omics” suggests a comprehensive assessment of numerous molecules [3]. Omics approaches offer a holistic view of the molecules that make up a cell or organism to identify genes (genomics), metabolites (metabolomics), mRNA (transcriptomics), and proteins (proteomics) in a non-biased biological context. Globally, web-based databases are an important resource for plant genomics, specifically detecting stress-reactive genes [4]. Functional genomics has helped to detect stress-related genes in crops [5,6]. Accessibility to the whole genome sequence of numerous plant species and recent developments in genomic approaches promise to deliver methods for locating stress-responsive genes at the genome-wide level. For complex trait loci, genome-wide association studies have identified stress-responsive genes and their favorable alleles. The advancement of genetic databases has enabled bioinformatics tools to identify stress-resistant gene families in various plant species using synteny and homology.

For targeted genome editing, three methods are currently available: transcription-activator-like effector nucleases (TALEN), clustered regularly interspaced short palindromic repeats (CRISPR), and zinc finger nuclease (ZFN). In cells, CRISPR-Cas9 is a cheap, easy, fast, and effective system for gene knockout [7]. For effective genome engineering, CRISPR-Cas9 has been used in animals, plants, and bacteria [8,9,10,11]. Furthermore, CRISPR-Cas9 has been used for high-throughput screening of genes, gene knockout, chromosomal loci live-cell labeling, endogenous gene expression, and single-stranded RNA (ssRNA) edition. The application of CRISPR-Cas9 for studying the function of a gene has generated disease models. However, several queries and challenges need to be addressed. CRISPR-Cas9 will likely enhance our comprehension of disease activity and its management. For targeted genome engineering, detecting programmable nucleases that produce cuts in double-strands has radically changed molecular biology; ZFNs pioneered this success, with TALEN extending the genome modifying capacity [12]. Globally, CRISPR-Cas9 received recognition from researchers for its visible benefits over TALEN and ZFN [13], being its (1) ease of designing target, (2) ability to create mutations by inserting the guided RNA and Cas9 protein, and (3) multiplexing ability to target several genes at one time [14,15]. Omics and CRISPR-Cas9 technology are poised to identify stress tolerance genes, molecular insight, and genome engineering to generate stress tolerance in crops. Developing and improving modern technologies to modify plant genomes and accumulate sufficiently large volumes of experimental molecular biological data will help create new schemes and approaches to improve economically valuable traits in plants and develop new varieties of important crops.

## 2. Multi-Omics Technology

### 2.1. Genomics

In plants, functional genomics has identified several genes that control abiotic and biotic stress reactions [16,17]. Some genes have been engineered to develop stress (biotic and abiotic) resistance in crop plants [18,19,20,21,22]. Numerous new candidate genes have been discovered from wild crop relative genomics for stress (abiotic and biotic) tolerance in crops [2,23]. For example, a high-density buckwheat complete genome sequencing genomic map, Hi-C online accessible sequencing data, and fosmid DNA libraries [17]. The authors also detected whole-genome duplication, identified numerous candidate genes for drought, cold stress, and heavy metal stress resistance, and predicted nearly 33,500 genes. Another study identified 33 transcription factors (TFs) of the tea plant using the transcriptomic and genomic database (http://planttfdb.cbi.pku.edu.cn/), which were classified into four groups (HD-Zip I to IV) after analyzing common motifs and domains [24]. A protein interaction was found. The results highlighted the diverse expression of *Cshdz* genes to salinity, drought, high and low temperature, and the association between *Cshdz* genes and resistant plants. In *Solanum americanum*, integrated RenSeq and genetic mapping were used to locate the genetic locus that confers resistance against late blight [25]. In wheat, MutRenSeq, a new version of RenSeq, was used to isolate R genes that confer resistance against stem rust [26]. Genome-wide analysis with ChIP-seq identified 21 ABA-associated TFs and their broad regulatory network [27]. Furthermore, a novel family of TFs was identified in *Arabidopsis* that was functionally involved in salt reactions and ABA. Genotyping by sequencing (GBS) is a newly discovered genomics technology for inspecting plant genetic diversity at a whole-genome level. An F2 population of *Brassica olearacea* was used to develop a high-density genetic map covering 879.9 cM, genotyped by 4103 single nucleotide polymorphisms (SNPs) [28]. The authors detected two major quantitative trait loci (QTLs) that confer resistance against clubroot resistance. The integration of high-throughput phenotyping and functional genomics delivers new approaches for crop improvement systems.

### 2.2. Transcriptomics

RNA profiling—realized recently using microarrays, gene expression, digital profiling, RNA sequencing, and serial analysis of gene expression [29]—can identify multiple stress resistance-related candidate genes, inferring relevant gene functions. The available online databases provide whole genome-wide transcriptomics data for plant stress reactions [30,31]. In *Arabidopsis*, transcriptomic analysis under drought and heat stress identified nearly 770 unchanged transcripts with 53 dissimilar specific proteins [32]. These findings were confirmed in sunflower [33]. Furthermore, combined heat and drought upregulated stress cytosolic ascorbate peroxidase1 (APX1) [34]. In chickpea, serial analysis of gene expression (SAGE) and next-generation sequencing (NGS) approaches were used to analyze the total transcriptome of drought- and salt-stressed plants [33,34]. Similarly, the subtractive cDNA suppression hybridization method was used in stressed chickpea plants [35]. A comparative microarray approach provided information on functional genes and pathways crosstalk in multiple stress transcriptomic studies in cotton [36]. In maize, RNA sequencing was performed to understand the adverse effects of cold, drought, salt stress, and heat stress [37]. Li et al. documented differentially expressed genes associated with signaling pathways, transcription, and metabolism [38]. RNA gel blot and microarray combined approaches verified that DREB2A, a transcription factor, controls the expression level of drought and cold stress genes [38]. Serial analysis of gene expression (SAGE) has been used extensively in plants to study gene-related responses against stresses. For example, in rice, from 5921 expressed genes, almost 10,122 tags were analyzed. Of 50,519 tags by global gene expression, 15,131 tags were similar to distinctive transcripts [39]. The integration of RNA-seq and bulked segregant analysis, called BSR-seq, has the power to enhance stress resistance in plants. For instance, *Bra019409* and *Bra019410* were possible candidate genes for clubroot resistance in *Brassica rapa* [40,41]. RNA-seq-mediated gene expression analysis could accelerate plant breeding by garnering knowledge on host-P interactions and identifying stress-related genes.

### 2.3. Proteomics

The qualitative and quantitative study of total proteins expressed in a cell, tissue, or organism is known as proteomics [42]. In the context of plant stress tolerance, entire proteomes are studied; however, numerous studies have investigated the cell wall proteome, organellar proteome, proteogenome, nuclear proteome, and phosphoproteome [43]. Several forms of mass spectrometry were used recently to profile the proteome in response to abiotic stresses [42,44,45]. Mass spectrometry for proteomics provides extensive proteome information when used in plant stress reactions and genome-wide studies. Proteome profiles can be compared to identify the function of particular proteins in biotic- and abiotic-induced stress signaling and differentially expressed stress-resistant proteins. Furthermore, phosphorylation group proteins play an important role in abiotic stresses [42,46]. A study on a proteome matrix in water-stressed rice identified signaling proteins and reactive oxygen species [47]. Various studies have used proteomics to highlight heavy metal stress in *Brassica juncea* [48], *Glycine max* [49], *Linum usitatissimum* [50], and *Arabidopsis thaliana* [51]. Heidarvand and Maali-Amiri (2013) comprehensively studied the proteomic profile of chickpea exposed to cold stress [52]. The phosphoproteome of wheat leaves has also been studied [53]. Several isoforms of *S*-adenosylmethionine in soybean under flooding and drought have been identified [54]. In tomato, signaling nuclear proteins with crosstalk chloroplast proteins were reported in drought-stressed plants [55]. Another study used tandem MS and two-dimensional gel electrophoresis (2-DE) approaches in waterlogged barley regimes to reveal the proteome profile [43]. The authors noted that sensitive barley genotypes had reduced photosynthetic performance and total biomass. Differentially expressed proteins in roots and leaves were associated with antioxidants and energy metabolism [43]. In *Eriobotrya japonica*, RNA-seq with isobaric tags relative absolute quantification (iTRAQ) was used to understand the cold tolerance mechanism [56]. The results revealed 1210 differentially expressed genes (DEGs) and 300 differentially expressed proteins (DEPs); of 3620 genes, only 27 shared both DEPs and DEGs. Kyoto encyclopedia of genes and genomes (KEGG) analysis predicted that biosynthesis of secondary metabolites and metabolic pathways were common. Real-time quantitative reverse transcription polymerase chain reaction (qRT-PCR) validation showed that gene expression of phenylalanine ammonia-lyase, anthocyanin synthase, and NADP-D-sorbitol-6-phosphate dehydrogenase was consistent with the transcriptome profile. Lou et al. suggested that these three genes play an important role in cold tolerance. Proteomics is a new technology for identifying proteins and pathways linked to the plant stress response and plant physiology. Moreover, proteomics enhances the understanding of stress-related proteins applied to molecular biology for crop improvement.

### 2.4. Metabolomics

Metabolomics is a high-throughput assessment of all metabolites in an organism. For exogenous and endogenous metabolites, scientists use non-targeted and targeted techniques [57]. Metabolites—including organic acids, peptides, secondary metabolites, steroids, hormones, ketones, vitamins, aldehydes, amino acids, and lipids—generate extensive data compared to transcriptomics and proteomics [58]. Advances in liquid chromatography–mass spectrometry (LC-MS), gas chromatography–mass spectrometry (GC-MS), direct injection mass spectrometry (DIMS), nuclear magnetic resonance (NMR), and high-performance liquid chromatography (HPLC) with other metabolomic approaches have further clarified stress tolerance processes and metabolite profiling [59]. There are almost 250,000 metabolites in plants; the concentration and total number are considerably higher in stressed than non-stressed environments [60]. The detection of valid metabolomic markers will enhance stress tolerance in plants [59,61]. Numerous researchers have documented metabolic profiles under stress environments in plants [62,63,64,65]. For example, drought-stressed *Arabidopsis thaliana* accumulated various metabolites containing proline, gamma-aminobutyrate (GABA), raffinose oligosaccharides, and others in the tricarboxylic acid cycle. Furthermore, activation of stress metabolic pathways and transcriptional regulation was dependent on abscisic acid (ABA) [66]. The superoxide dismutase gene was engineered into *Populus* plants, and data processing generated information on reactive oxygen species (ROS) metabolism [67]. Feng et al. (2013) reported a reduction in glycolysis-related sugar levels in salt-stressed barley leaves [68]. Shen et al. (2016) studied drought stress in chickpea varieties, which increased branched-chain amino acids and allantoin and decreased glucosamine, aspartic acid, and aromatic amino acids [8]. In *Arabidopsis*, transcription factor genes, *Myb28* and *Myb29*, particularly for aliphatic GSL production and biosynthetic gene expression, unknown genes, and regulatory networks were estimated by integrating metabolic profiling and transcriptome data [69]. Furthermore, overexpression of these TFs in *Arabidopsis* produced industrial GSLs. Functional genomics, metabolomics, transcriptomics, and proteomics open a new direction for decoding secondary metabolism. We suggest that omics approaches from multiple platforms could provide molecular insight and enhance stress resistance through plant breeding. Table 1 summarizes some available databases and their URLs.

## 3. CRISPR Technology

Due to its robust success, CRISPR-Cas9 is becoming a potential tool for genetically enhancing desirable crop traits, i.e., disease resistance, nutrient content, adaptation to multiple stresses, plant architecture, and yield. In some cases, a specific trait can be improved by negative regulatory gene knockout. Rice grain weight improved with gene modification of some QTL [70]. Maize grain yield under drought increased with genome engineering of the ARGOS8 locus [71]. In woody plants, CRISPR-Cas9 produced mutants in the first transgenic generation; this is significant as woody plant breeding is difficult due to their long lifespan [72,73]. Another study knocked out the *OsGAN1* gene in rice and verified that it regulates root length and plant height [74]. Similarly, *OsABCG26* gene knockout verified that this gene regulates pollen exine and anther cuticle, and *OsTCD10* had a substantial role in chloroplasts of cold-stressed rice [75,76]. Figure 1 summarizes the principles of CRISPR-Cas9.

### 3.1. CRISPR-Cas9 Genome Engineering to Biotic Stress Tolerance

Genome editing by CRISPR-Cas9 has been used effectively in several crops, including cotton, maize, rice, and wheat. However, most genome engineering studies have targeted biotic stresses, such as diseases. In wheat, the CRISPR-Cas9 method was used successfully to knock out all three *EDR1* homologs to create plants (Taedr1) with increased tolerance to powdery mildew [77]. In *Arabidopsis*, the knockout of susceptible gene *EDR1* increased resistance to powdery mildew [78]. Recessive resistance genes, *eIF* (eukaryotic translation initiation factor), have been detected in several dissimilar hosts, with *eIF* (iso) *4E* and *eIF4E* genes used with CRISPR-Cas9 to form virus-resistant plants in *Arabidopsis* and cucumber, respectively [79,80]. *CsLOB1* is a susceptible gene of the citrus canker (causative agent; *Xanthomonascitri*); CRISPR-Cas9 was used to edit this gene to develop resistant grapefruit plants [81,82]. Additionally, a negative resistance function *MLO* gene, responsible for powdery mildew susceptibility, was mutated successfully by Cas9 knockouts to enhance resistance against powdery mildew in tomato and wheat [83,84,85]. The application of CRISPR-Cas9 as an antivirus tool cleaved beet severe curly top virus, which decreased the viral infection [86,87]. The rice tungro spherical virus (RTSV), linked to the negatively controlled susceptible *eIF4G* gene, was eliminated using CRISPR-Cas9 to develop resistant rice varieties [88]. From CRISPR-Cas9, the loss of function *VvWRKY52* gene produced resistance against *Botrytis cinerea* in grape (*Vitis vinifera*) [89]. Furthermore, CRISPR-Cas9 has been used to interrupt multiple virus genomes, including CLCuK_0_V, TYLCSV, and TYLCV [90]. For cucumber mosaic virus and tobacco mosaic virus, a technology to modify RNA virus genomes has been advanced from sgRNA and FnCas9. Hence, molecular immunity to RNA viruses was mediated by sgRNA/FnCas9 expression in *Arabidopsis* and tobacco [91]. CRISPR-Cas9 successfully targeted *OsERF922* against blast fungus resistance in rice [92]. Plant ethylene-responsive factors (ERFs) can control tolerance against various stresses because they are involved in the ethylene (cytokinin) pathway [93]. When taken together, these reports deliver robust indications that CRISPR-Cas9 can enhance biotic stress resistance in plants. Figure 2 summarizes omics and CRISPR-Cas9 strategies for stress-tolerant crop production.

### 3.2. CRISPR-Cas9 Genome Engineering to Abiotic Stress Tolerance

Abiotic stress tolerance mediated by various genes is a complex trait. There are major interactions and crosstalk among components of metabolic, regulatory, and signaling pathways [94,95]. CRISPR-Cas9-mediated genome editing can be used to modify almost any sequence (depending on accessibility to the protospacer adjacent motif, PAM site) to reveal its function in the genome. Molecular breeders have discovered numerous abiotic-stress-resistant T genes and engineered them into crop plants. CRISPR-Cas9-generated mitogen-activated protein kinases3 (slmapk3) gene mutants increased the defense response to drought in tomato (*Solanum lycopersicum*) [96]. CRISPR-Cas9 was used to generate mutants in rice to understand the mechanism of stress-ABA-activated protein kinase2 [97]. In *Arabidopsis* under cold stress, CRISPR-Cas9 was used to generate mutants (cbfs double and triple mutants) to determine the role of C-repeat binding factors [98]. In maize, the CRISPR-Cas9 approach was used to increase the expression level of the ARGOS8 gene (negatively regulate ethylene response) to develop drought tolerance; the promoter of ARGOS8 changed into GOS2. These mutants had enhanced grain yields under drought conditions in the field [69]. Moreover, overexpressing *TaCP* and *SPCP2* increased drought tolerance in *Arabidopsis* [99,100,101]. Plants overexpressing the melatonin biosynthesis genes were identified as abiotic stress-tolerant [102,103]. In hybrid rice, targeted editing of the TMS5 gene led to the rapid formation of temperature-sensitive breeding lines [104]. Plant breeding activities may have reduced T gene alleles after selecting yield-related genes during domestication programs [105]. Breeders have developed stress-tolerant crops with gene function information [3]. The above examples show that CRISPR-Cas9 can modify/eliminate genes to mediate resistance against numerous abiotic stresses, e.g., salinity, drought, extreme temperatures, heavy metals, and nutrient deficiencies [106,107] (Table 2).

## 4. Conclusions and Perspectives

High-throughput verification of experimental platforms is required to reveal gene functions in plants. Innovative efforts include plant phenotyping (http://www.lemnatec.com), metabolomics, and enzyme assay (http://www.biolog.com) platforms [113]. However, the prediction of gene function based on networks is an active research area but limited in plant science. We need additional data, easy access to tools and data, improved data analysis, and high-throughput verification from experiments to achieve a network-based gene function identification goal. Omics technologies, databases, and bioinformatics tools primarily provide information on candidate genes, biosynthetic pathways, proteins, master regulators, biological networks, and cross talk, especially on the plant stress response.

The CRISPR-Cas9-mediated genome editing system has fundamentally influenced gene function research and, ultimately, crop improvement [114,115]. The plant genome engineering approach has no ethical issues. CRISPR-Cas9 mutants are generated with greater efficiency and specificity than TALEN and ZFN. Hence, the CRISPR-Cas9-mediated genome editing system has great potential for practical research. Various CRISPR-Cas9 platforms have been developed for plant genome engineering but require advanced targets for specificity and efficiency.

Moreover, gene replacement and DNA part knock-in is a challenge [116]. Cas9 variants, gene repression, and activation domains can control target gene expression [117]. Hence, this system could be used to develop climate-resilient crops.

A toolbox based on CRISPR-Cas9 has been established for gene repression and activation in plants [118]. CRISPR-Cas9 could be adapted for new approaches, e.g., epigenomic regulation, chromatin imaging, and RNA cleavage [40,41,119]. Given its versatility, simplicity, efficiency, and flexibility, the future of functional genomics is likely to depend on the CRISPR-Cas9 system. Omics and CRISPR have provided a snapshot for improving an organism’s functioning and interactions at the cell and tissue level by depicting and measuring biomolecules.

## Figures and Tables

**Figure 1 ijms-22-01292-f001:**
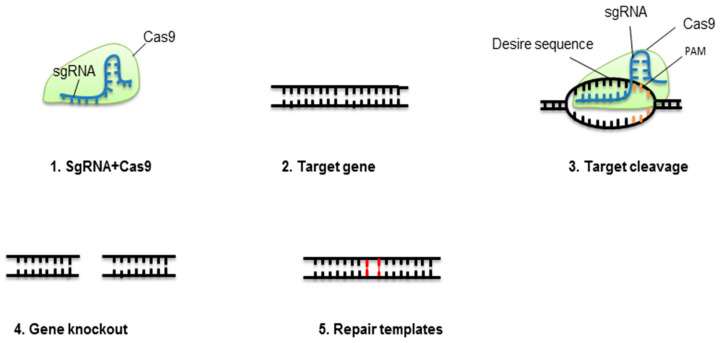
Concept of CRISPR-Cas9-mediated gene elimination. Single guide RNA (sgRNA) containing crRNA and tracrRNA fixes to Cas9 protein. This complex will break at a specific target of the double-stranded DNA molecule. The nonhomologous end-joining pathway (NHEJ) will repair the cleaved location.

**Figure 2 ijms-22-01292-f002:**
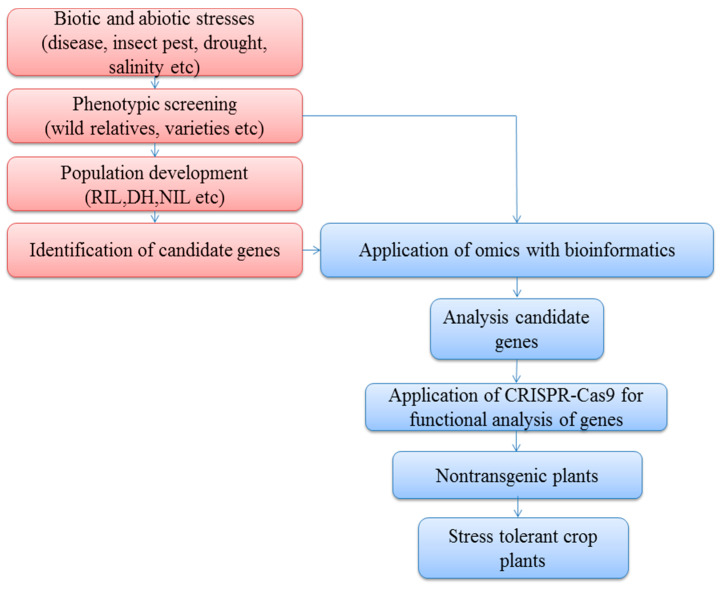
Omics and CRISPR-Cas9 strategies for garnering knowledge on molecular systems and gene function as the main objective for producing stress-tolerant crop plants. RIL: Recombinant inbred line, DH: Doubled haploid, NIL: Near isogenic line.

**Table 1 ijms-22-01292-t001:** Accessible genome level databases.

Name	Species	Database Resource	URL
TAIR	Mainly for *Arabidopsis thaliana*	Whole genome	http://www.arabidopsis.org
1001genomes	*Arabidopsis thaliana*	Whole genome	http://www.1001genomes.org
Phytozome	Numerous	Whole genome	http://www.phytozome.net
NCBI	Numerous	Whole genome	http://www.ncbi.nlm.nih.gov
Cottongen	*Gossypium* spp.	Whole genome and breeding	http://www.cottongen.org
Soybean breeders toolbox	*Glycine max*	Whole genome	http://www.soybase.org
MaizeGDB	*Zea mays*	Whole genome	http://www.maizegdb.org
RAP-DB	*Oryza sativa*	Whole genome	http://rapdb.dna.affrc.go.jp
PlantGDB	Numerous	Whole genome	http://www.plantgdb.org
IWGSC	*Triticum aestivum*	Whole genome	http://www.wheatgenome.org
Gramene	Numerous	Whole genome	http://www.gramene.org
Ensemblplants	Numerous	Whole genome	http://plants.ensembl.org
KEGG	Numerous	Whole genome	http://www.genome.jp/kegg/genome/plant.html
Graingenes	Numerous	Whole genome	http://wheat.pw.usda.gov/GG2/index.shtm
PMN	Numerous	Metabolomics	http://www.plantcyc.org
CSB.DB	*Arabidopsis thaliana*	Metabolomics	http://csbdb.mpimp-golm.mpg.de/csbdb/gmd/gmd.html
PRIMe	*Arabidopsis thaliana*	Metabolomics	http://prime.psc.riken.jp/lcms/ms2tview/ms2tview.html
AFGN	*Arabidopsis thaliana*	Gene expression	https://www.deutsche-botanische-gesellschaft.de/en/about-us/afgn
OryzaExpress	*Oryza sativa*	Gene expression	http://plantomics.mind.meiji.ac.jp/OryzaExpress/
RGAP	*Oryza* spp.	Gene expression	http://rice.plantbiology.msu.edu
CottonFGD	*Gossypium* spp.	Gene expression	http://www.cottonfgd.org
Genevestigator	Numerous	Gene expression	http://genevestigator.com
TriFLDB	*Triticum aestivum*	Gene expression	https://bigd.big.ac.cn/databasecommons/database/id/3452
BAR	Numerous	Gene expression	http://bar.utoronto.ca/welcome.htm
NOBLE	*Medicago truncatula*	Gene expression	http://mtgea.noble.org/v2
Uniprot	Numerous	Proteomics	http://www.uniprot.org/proteomes/
RICE PROTEOME	*Oryza sativa*	Proteomics	http://gene64.dna.affrc.go.jp/RPD
Proteomics database	*Arabidopsis thaliana*	Proteomics	http://proteomics.arabidopsis.info
SUBA	*Arabidopsis thaliana*	Proteomics	http://www.suba.bcs.uwa.edu.au/
AGRIS	*Arabidopsis thaliana*	Transcription factor	http://arabidopsis.med.ohio-state.edu
PlantTFDB	Numerous	Transcription factor	http://planttfdb.gao-lab.org/
LegumeTFDB	*Lotus japonicas*, *Medicago truncatula*, *Glycine max*	Transcription factor	http://legumetfdb.psc.riken.jp
Grassius	*Zea mays*, *Oryza sativa*, *Sorghum bicolor*	Transcription factor	http://grassius.org/
TRIM	*Oryza sativa*	Mutants	http://trim.sinica.edu.tw
RMD	*Oryza* spp.	Mutants	http://rmd.ncpgr.cn/
ABRC	*Arabidopsis thaliana*	Mutants	http://abrc.osu.edu
NASC	*Arabidopsis thaliana*	Mutants	http://arabidopsis.org.uk/home.html
Fox Hunting	Numerous	Mutants	http://nazunafox.psc.database.riken.jp
SIGnAL	*Arabidopsis thaliana*	Mutants	http://signal.salk.edu

**Table 2 ijms-22-01292-t002:** CRISPR-Cas9 application for crop improvement.

Species	Traits	Target Genes	Reference
*Abiotic stresses*			
Rice	Improved resistance to arsenic stress	*ARM1*	[108]
	Depletion of Cd into grain	*LCT1*	[109]
	Depletion of Cd into grain	*Nramp5*	[107]
	Drought tolerance	*SAPK2*	[97]
Tomato	Drought tolerance	*SIMAPK3*	[96]
Maize	Drought tolerance	*ARGOS8*	[69]
*Arabidopsis*	Cold tolerance	*CBF1 CBF2*	[99]
*Biotic stresses*			
*Arabidopsis*	Resistance to turnip mosaic virus	*eIF (iso)4E*	[80]
Wheat	Improved resistance to powdery mildew	*TaMLO*	[85]
	Improved resistance to powdery mildew	*EDR1*	[77]
Rice	Increased resistance to blast fungus	*OsERF922*	[92]
	Increased resistance to tungro spherical virus	*eIF4G*	[88]
Barley	Improved resistance to fungal pathogens	*MORC1*	[110]
Orange	Improved resistance to citrus canker	*CsLOB1*	[111]
Tomato	Improved resistance to powdery mildew	*Mlo1*	[84]
	Anthocyanin biosynthesis	*ANT1*	[112]
Grape	Improved resistance to *Botrytis cinerea*	*WRKY52*	[89]
Cucumber	Virus resistance	*eIF4E*	[79]

## Data Availability

Not applicable.

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
