# Peer review of "Omics and CRISPR-Cas9 Approaches for Molecular Insight, Functional Gene Analysis, and Stress Tolerance Development in Crops"

_ijms, 2021, doi:10.3390/ijms22031292_

Round 1

Reviewer 1 Report

Overall I expect the manuscript gives knowledge on integration of OMICS and CRISPR-Cas9 on improved tolerance. However, except CRISPR-Cas9 part on biotic stress most of the subsections are very general.

I suggest author to choose few plants which are well studied using OMICS and CRISPR-Cas9 techniques. Then give us some informative view on how the techniques helps to understand and solve the problem like stresses.

Especially, Author must address and focus writing on problems solved by CRISPR-Cas9 techniques and usage of genomics data.

Through out this manuscript, author is giving definitions such as Systems biology, Genomics, transcriptomics, proteomics, metabolomics. Like these statements are good for book chapters that too if it is deal with subject. For here, I felt it is not needed.

CRISPR-Cas9 on biotic stress is informative and written well.

All the best.

P.S: I have given more comments on the manuscript file. Kindly find the attachment.

Author Response

Point 1: Overall, I expect the manuscript gives knowledge on integration of OMICS and CRISPR-Cas9 on improved tolerance. However, except CRISPR-Cas9 part on biotic stress most of the subsections are very general.

Response 1: We thank the reviewer for their positive comment.

Point 2: I suggest author to choose few plants, which are well studied using OMICS and CRISPR-Cas9 techniques. Then give us some informative view on how the techniques helps to understand and solve the problem like stresses.

Response 2: We agree with this valid concern; for this reason, we have added Table 1 to address the above issue.

Point 3: Especially, Author must address and focus writing on problems solved by CRISPR-Cas9 techniques and usage of genomics data.

Response 3: The problems solved by CRISPR-Cas9 techniques are in Table 2. Functional genomics, especially metabolomics, transcriptomics, and proteomics substantially contributed to plant molecular responses against stresses. Functional genomics data is delivering new ways for crop improvement systems (lines 90–216).

Point 4: Throughout this manuscript, author is giving definitions such as Systems biology, Genomics, transcriptomics, proteomics, metabolomics. Like these statements are good for book chapters that too if it is deal with subject. For here, I felt it is not needed.

Response 4: We have deleted the above definitions.

Point 5: CRISPR-Cas9 on biotic stress is informative and written well.

Response 5:  We thank the reviewer for their positive comment.

Point 6: I have given more comments on the manuscript file. Kindly find the attachment.

Response 6: We have noted all the comments in the manuscript and relevant changes have been made in the revised manuscript.

Reviewer 2 Report

This review paper is timely and should be useful for the readers. 

I enjoyed reading this paper and learned. 

I found only several technical mistakes at citation, 

and several in-consistency.  

These kind of mistake could be checked by technical editors,

but I checked in yellow line marker.

Please check and revise.  

Author Response

Point 1: This review paper is timely and should be useful for the readers. I enjoyed reading this paper and learned.

Response 1: We thank the reviewer for their positive comment.

Point 2: I found only several technical mistakes at citation, and several in-consistencies. These kind of mistake could be checked by technical editors, but I checked in yellow line marker. Please check and revise.

Response 2: Apologies. All typos have been corrected in the revised manuscript.

Reviewer 3 Report

The manuscript by Razzaq et al. summarized the omics and CRISPR technologies application of crop stress tolerance. This topic is very interesting and the content is helpful for us to understand this application in crop stress tolerance. However, there were some issues for the presentation of current manuscript. The logic of whole manuscript was chaos and very hard to follow. I suggested the authors present the manuscript as following, The multi-omics technology, CRISPR technology, multi-omics application in identifying crop stress tolerance related potential genes, CRISPR application in confirm crop stress tolerance related gene, function and crop breeding. For current manuscript, the authors just described the findings from previous studies with very few own summaries and comments. There were lots of grammar errors, please improve.

Author Response

Point 1: The manuscript by Razzaq et al. summarized the omics and CRISPR technologies application of crop stress tolerance. This topic is very interesting and the content is helpful for us to understand this application in crop stress tolerance. However, there were some issues for the presentation of current manuscript. The logic of whole manuscript was chaos and very hard to follow. I suggested the authors present the manuscript as following, The multi-omics technology, CRISPR technology, multi-omics application in identifying crop stress tolerance related potential genes, CRISPR application in confirm crop stress tolerance related gene, function and crop breeding. For current manuscript, the authors just described the findings from previous studies with very few own summaries and comments. There were lots of grammar errors, please improve.

Response 1: We thank the reviewer for the encouraging comments. We have incorporated the changes (logic of manuscript) in the revised manuscript. We have also checked the grammar and improved it in the revised manuscript.

Round 2

Reviewer 1 Report

Dear Author, 

        I can see you have made several changes. I appreciate it. However, in section 4, except subsection 4.2, rest of the manuscript are merely descriptive. I understand that in review manuscript you need to give some definitions. But those are for some general topics. Here you have chosen the omics and CRISPR-Cas9 which are cutting edge technology and using for several applications including crop improvement. So you can more information to give.

If you already covered your topic then no need extend the manuscipt with so many descriptions. I don't find subsection 4 is informative. Kindly remove that. It author thinks this manuscript will be too short if you remove some subsection, I suggest to explain trait improved two plants with detailed mechanism of how OMICS and CRISPR-Cas9 technology were utilized. Not just as the Table 1. Also there is some concern in Table 1.

         This manuscript needs to be intensly revised again to remove redundancy and description.

         Send the manuscript with "track change" option only for insertions. Marked deletions make difficulty in reading. More comments can be find in manuscript file.

All the best!          

Author Response

Point 1: I can see you have made several changes. I appreciate it.

However, in section 4, except subsection 4.2, rest of the manuscript are merely descriptive. I understand that in review manuscript you need to give some definitions. But those are for some general topics. Here you have chosen the omics and CRISPR-Cas9 which are cutting edge technology and using for several applications including crop improvement. So you can more information to give. If you already covered your topic then no need extend the manuscipt with so many descriptions. I don't find subsection 4 is informative. Kindly remove that. If author thinks this manuscript will be too short if you remove some subsection, I suggest to explain trait improved two plants with detailed mechanism of how OMICS and CRISPR-Cas9 technology were utilized. Not just as the Table 1. Also there is some concern in Table 1. This manuscript needs to be intensely revised again to remove redundancy and description. Send the manuscript with "track change" option only for insertions. Marked deletions make difficulty in reading. More comments can be find in manuscript file.

Response : We thank the reviewer for his/her positive comments. We have changed the title of manuscript from integration of omics and CRISPR to Omics and CRISPR. We have deleted section 4 (except subsection 4.2) from the manuscript.  We have noted the annotations in the manuscript and made relevant changes in the revised manuscript and Table 1. We hope that the revised manuscript is acceptable for publication.

Reviewer 3 Report

The authors have addressed my concerns

Author Response

We could not find any written comments from Reviewer three, however all annotations on the manuscript have been revised.